# Delayed Reaction of Radiation on the Central Nervous System and Bone System in C57BL/6J Mice

**DOI:** 10.3390/ijms25010337

**Published:** 2023-12-26

**Authors:** Yi Ru, Xianxie Zhang, Baoying Shen, Chunqi Yang, Huijing Yu, Zuoxu Liu, Xiangjun Wu, Fangyang Li, Jialu Cui, Chengcai Lai, Yuguang Wang, Yue Gao

**Affiliations:** Department of Pharmaceutical Sciences, Beijing Institute of Radiation Medicine, Beijing 100850, China; ruyi9809@163.com (Y.R.); zhangxianxie@163.com (X.Z.); shenby25511@163.com (B.S.); ycq1qaz@outlook.com (C.Y.); 1566247926@sina.cn (H.Y.); m15666072026@163.com (Z.L.); xiangjunwu2022@163.com (X.W.); lifangyang7012@163.com (F.L.); cuijialu143021@163.com (J.C.); asa2057516@163.com (C.L.)

**Keywords:** radiation injury, ionizing radiation, delayed radiation damage, brain injury, bone loss

## Abstract

The aim of this study was to provide a suitable mouse model of radiation-induced delayed reaction and identify potential targets for drug development related to the prevention and treatment of radiation injury. C57BL/6J mice were subjected to singular (109 cGy/min, 5 Gy*1) and fractional (109 cGy/min, 5 Gy*2) total body irradiation. The behavior and activity of mice were assessed 60 days after ionizing radiation (IR) exposure. After that, the pathological changes and mechanism of the mouse brain and femoral tissues were observed by HE, Nissl, Trap staining micro-CT scanning and RNA sequencing (RNA-Seq), and Western blot. The results show that singular or fractional IR exposure led to a decrease in spatial memory ability and activity in mice, and the cognitive and motor functions gradually recovered after singular 5 Gy IR in a time-dependent manner, while the fractional 10 Gy IR group could not recover. The decrease in bone density due to the increase in osteoclast number may be relative to the down-regulation of RUNX2, sclerostin, and beta-catenin. Meanwhile, the brain injury caused by IR exposure is mainly linked to the down-regulation of BNDF and Tau. IR exposure leads to memory impairment, reduced activity, and self-recovery, which are associated with time and dose. The mechanism of cognitive and activity damage was mainly related to oxidative stress and apoptosis induced by DNA damage. The damage caused by fractional 10 Gy TBI is relatively stable and can be used as a stable multi-organ injury model for radiation mechanism research and anti-radiation medicine screening.

## 1. Introduction

Ionizing radiation (IR) is widely used in daily life (e.g., energy generation), medicine (e.g., chemotherapy and radiation therapy), and other fields [1,2,3]. With its wide application, IR can cause harmful effects on human health. Radiation-induced damage is never localized to a single organ and produces a range of subsequent adverse effects, which is often the greatest concern after radiotherapy and accidental irradiation: people tend to be concerned about whether they have suffered damage to a healthy organ or whether they have a delayed effect long after irradiation that requires pharmacological intervention. In recent years, researchers have conducted extensive studies on the damage caused by IR, and there is now a large number of studies showing that acute exposure to IR can cause serious damage to the gastrointestinal system, immune system, and hematopoietic system [4,5,6,7].

In previous studies, high-dose radiation was paid more attention, while adverse reactions after radiotherapy do not necessarily appear immediately in the short term and were ignored. Some of them may appear one month, two months, or even years after radiotherapy, which is known as a delayed reaction. There is no clear definition of delayed reaction, but there are lots of clinical cases of delayed reactions after radiotherapy, which pose many new challenges to the quality of life of cancer patients [8]. Some of the common delayed reactions include radiation-associated caries injury, post-radiation-induced head drop syndrome involving neck extensor muscle weakness reactions, and radiation cerebral delayed reactions [9,10]. However, the exact mechanism, such as senescence related to delayed reactions and the specific radiation dose, is not clear.

In addition to the serious damage to the systems above, IR can also cause some damage to the central nervous system, mainly because of its high sensitivity to IR exposure. The amount of literature suggests that post-operative radiotherapy in cancer patients can cause some degree of cognitive impairment [11,12]. Some clinical studies of radiation-induced brain injury symptoms may be acute [13,14], sub-acute, or chronic, occurring hours, days, weeks, months, or even years after IR exposure, and yet children appear to be more likely than adults to suffer age-related deficits in neurocognitive skills [15,16]. The literature suggests that the pathogenesis is mainly due to oxidative stress and inflammation, indicating neurological damage caused by radiation [17]. In addition, radiation-induced cognitive impairment is mainly related to the time of damage after radiation exposure and hippocampal-dependent memory deficits, and the exact evidence is still not confirmed.

Bone atrophy and increased risk of bone fracture are consequences of exposure to radiation for cancer treatment [18,19]. Osteopenia and osteoporosis have been characterized as pathological conditions after therapeutic irradiation. The main causes of bone loss after radiotherapy, on the one hand, are the damage to osteoblasts and bone vessels themselves. On the other hand, there is an abnormal function of osteoclasts, which leads to the injury of bone metabolism. Increased evidence shows that bone loss in mice after total body irradiation is mainly due to the maturation and differentiation of osteoclasts induced by ionizing radiation (IR) exposure, which significantly increases the number of osteoclasts in cancellous bone tissue and accumulates bone loss [20,21,22]. The specific mechanism may be related to iron overloading [23]. Despite evidence that bone loss will occur shortly after IR exposure, the current research is mainly focused on single IR exposure and short-term observation below 5 Gy, while there are few reports on the long-term response of bone loss caused by ionizing radiation.

The aim of this study was to investigate the effects of singular or fractional radiation exposures on cognitive function and activity in mice after 60 days and to establish an animal model of delayed radiation damage. Here, relevant behavioral tests, pathological analyses, transcriptome sequencing analyses, and molecular biological validation were performed to assess the cognitive and activity decline in mice due to singular or fractional IR exposures in these aspects.

## 2. Results

### 2.1. Effects of Singular and Fractional IR Exposure on Physiological Characteristics in Mice

The harm of radiation to human health has been widely discussed by people; however, current research is mainly focused on short-term radiation injury or acute radiation injury. Few studies pay attention to the delayed reaction caused by radiation. Therefore, we focus on the physical function damage in mice 2 months after singular or fractional IR exposure. As the most intuitive index, the physical characteristics and daily activity of mice were observed and evaluated at first. C57BL/6J mice were adaptively reared for five days prior to the establishment of the radiation injury model. To assess the long-term effect after IR exposure, the physical characteristics of mice were observed 60 days following IR exposure. The singular IR exposure group has a relatively minor alteration in appearance. Meanwhile, fractional IR exposure resulted in more severely pronounced hair graying and hair loss (Figure 1A). We compared the differences in activity and hair change after IR exposure (Figure 1B). In addition, the weight of mice was recorded weekly for eight weeks. Compared with the control group, both the IR exposure groups exhibited a significant decrease in weight during the second week after exposure to radiation. However, no significant difference was observed between the mice in the singular 5 Gy IR exposure group and the control group after the sixth week. Notably, the mice in the fractional 10 Gy IR exposure group experienced a significant decrease in weight and low body weight growth rate after being exposed to a second irradiation at 30 days, which differed significantly from the control group (Figure 1C). As a result, mice undergoing TBI exhibited anomalous physiological characteristics.

### 2.2. Singular and Fractional IR Exposure Leads to Decreased Activity and Spatial Memory in Mice

Through the physiological indicators, we preliminarily judged that two months later, IR exposure still caused damage to the physiological state of mice, but the specific type of injury is still unknown.

In this work, we did the OFT and MWM tests to explore the effect of IR exposure on cognitive function in mice. The OFT was conducted to investigate the effects of IR exposure on spontaneous exploratory ability and motor speed in mice. According to the result of the OFT test, these behavioral effects included decreased activity in the open field and the time spent in the central area (Figure 2A). As can be seen from the orbits of exploring (Figure 2B), both IR exposure groups, especially fractional 10 Gy IR exposure mice, moved mostly along the edges and less into the central area, showing depression-like behavior.

MWM tests were performed to detect the IR-affected changes in spatial memory ability. During the hidden sessions, all treated groups improved their performance with training. Compared to the control group, both of the IR exposure groups displayed increased latency to the platform. In the probe trial, there was a significant difference in swimming speed between the control group and the fractional 10 Gy IR-exposed group; however, there is no significant difference in the other two parts (Figure 2C).

According to our findings, both singular 5 Gy and fractional 10 Gy IR exposure groups resulted in reduced activity levels and decreased spatial memory in mice, with the damage being more pronounced in the fractional 10 Gy IR exposure group.

### 2.3. IR Exposure Induced Brain Tissue Damage in Mice

From the results of behavioral experiments, it can be concluded that both IR exposure groups can lead to cognitive impairment in mice, and the cumulative 10 Gy damage was more obvious. Therefore, we further explored how IR exposure caused damage to mouse brain tissue. We performed HE staining and Nissl staining on mouse hippocampal tissue. HE staining mainly represents the pathological changes in mouse brain structure, and Nissl staining represents the changes in neuronal apoptosis in mouse brain tissue. The dentate gyrus (DG) of the hippocampus represents one of the main brain sites of adult neurogenesis. It has been proposed that radiation injury to the neural stem cell compartment of the hippocampus may be involved in cognitive decline [24,25]. The results of HE staining showed that no significant pathological changes were observed between the singular 5 Gy IR-exposed group and the control group. However, the fractional 10 Gy IR exposure group was observed to have varying degrees of abnormalities, such as loosely arranged pyramidal cells, crinkled neurons, deepened cell staining, and poorly demarcated nuclei in the DG region (Figure 3A). According to Nissl staining, both IR exposure groups showed swelling of neuronal cells, neuronal necrosis, and nucleolytic in the DG region, especially in the fractional 10 Gy IR exposure group (Figure 3B).

In conclusion, the fractional 10 Gy IR-exposed group showed more serious damage to cognitive function and neurons, as well as a certain degree of reduction in motor ability.

### 2.4. Transcriptome Sequencing Analysis of Mouse Hippocampal Tissue after IR Exposure

From the results above, we determined that the long-term effects of IR exposure still cause damage to the brain tissue of mice. To extend our knowledge of the genetic level of the long-term effects of IR exposed in the brain tissue, the hippocampus from the CON group, singular 5 Gy IR group, and fractional 10 Gy IR group were isolated for RNA extraction and high-throughput sequencing on the Illumina platform.

Cluster analysis of the samples based on DEGs showed that the two IR exposure groups had better gene separation compared to the control group. There were a total of 540 DEGs in the singular 5 Gy IR exposure group compared to the control group, of which 252 genes were expressed up-regulated, 288 genes were expressed down-regulated (Figure 4A), and a total of 438 DEGs in the fractional 10 Gy IR exposure group compared to the control group, of which 138 genes were expressed up-regulated, and 300 genes were expressed down-regulated (Figure 4B).

The number of DEGs in the control group, the singular 5 Gy IR exposure group, and the fractional 10 Gy IR exposure group were plotted on a Venn diagram (Figure 4C) to find the overlap DEGs in the CON-singular 5 Gy IR exposure group and CON-fractional 10 Gy IR exposure group. The result shows that there were 226 DEGs that may be important for the cognitive decline in mice caused by radiation.

The GO enrichment results of 226 DEGs in terms of down-regulated genes in the singular 5 Gy exposure group are mainly related to the regulation of calcium ion transport, cell fate commitment, cellular response to cytokine stimulus, tight junction organization, and phospholipid metabolic process. In the fractional 10 Gy IR exposure group, the biological progress was associated with the response to cAMP, regulation of calcium ion transport, and negative regulation of neuron apoptotic process. The cross-section of the two groups was counted, and it was found that 60 days after IR exposure, these biologically functionally relevant genes, including cerebrospinal fluid secretion, neuron differentiation, midbrain dopaminergic neuron differentiation, central nervous system neuron differentiation, nucleoside diphosphate phosphorylation, will still be down-regulated. In terms of molecular function, DGEs of both IR exposure groups were associated with DNA function, which included sequence-specific double-stranded DNA binding, aryl sulfotransferase activity, sequence-specific DNA binding, and scaffold protein binding. In terms of cellular composition, they are associated with the extracellular matrix in the extracellular space.

In contrast, up-regulated genes are mainly associated with immune functions in terms of biological processes, which include chemokine-mediated signaling pathway, signal transduction, cellular response to cholesterol, angiogenesis, positive regulation of calcium ion import, positive regulation of monocyte chemotaxis and cell–cell signaling in singular 5 Gy IR exposure group. In the fractional 10 Gy IR exposure group, the biological progress was associated with inflammatory response, lymphocyte chemotaxis, eosinophil chemotaxis, and positive regulation of inflammatory response. The up-regulated DEGs between the two IR exposure groups were mainly related to the biological processes of monocyte chemotaxis, immune response, and biological processes involved in interspecies interaction between organisms (Figure 4D).

Based on the results of the transcriptome, both singular and fractional IR exposures lead to the down-regulation of neuron-related gene expression and the up-regulation of immune response-related genes. Based on that, we focus our subsequent exploration of radiation-induced cognitive impairment on the following areas: immune response, neurotransmitter secretion, and neuronal differentiation.

### 2.5. The Effect of IR Exposure on DNA Damage and Cognition-Related Marks in Mouse Hippocampal Tissue

We explored possible mechanisms of radiation-induced delayed impairment of cognitive function using transcripts per million transcripts (TPM) to quantify the expression of characteristic genes.

IR directly causes DNA damage, and we quantified DNA damage marker-related genes (γH2AX), which were significantly elevated in both IR exposure groups compared to the control group and particularly significant in the fractional 10 Gy IR exposure group, which was also verified in the protein level by WB experiments (Figure 5A,B).

Also, TPM quantification was performed regarding cognition-related genes (BDNF, Tau). As shown in Figure 5C, BDNF and Tau genes were significantly reduced in both IR exposure groups and had significant differences in the control group. The results showed that compared to the control group, BDNF expression was reduced in both IR groups, which is consistent with the results of genetic statistics (Figure 5D). In our brain, Tau participates in microtubule formation and maintains microtubule stability, while abnormally phosphorylated Tau protein loses its ability to bind to tubulin, which leads to cognitive impairment [26,27]. The results showed that compared to the control group, p-Tau expression was elevated in both IR groups, suggesting that IR induced hyper-phosphorylation of Tau protein (Figure 5E).

### 2.6. IR Exposure Increases Oxidative Stress and Apoptotic Response in Mouse Hippocampal Tissue

Radiation-induced damage to the body is the result of a combination of pathways. The literature has highlighted that radiation causes oxidative stress and DNA double-strand breaks in the body, leading to a series of subsequent damages. In order to explore the mechanism of IR exposure-induced brain damage, we also quantified genes related to oxidative stress (p21, Nrf2, and HO-1) and apoptosis-related genes (bax, bcl2, and caspase3).

The Kelch-like ECH-associated protein 1- (Keap1-) nuclear factor-erythroid 2-related factor 2 (Nrf2) system can be used to monitor oxidative stress; Keap1-Nrf2 is closely associated with radiation and controls the transcription of multiple antioxidant enzymes [28,29]. After IR exposure, compared with the control group, both IR groups show down-regulation of Nrf2, HO-1, and Keap-1 and up-regulation of p21 (Figure 6A). The results verified protein level and showed a reduction in the expression of oxidative stress-related proteins (Nrf2 and HO-1) in both IR groups compared to the control group. Interestingly, according to the results of WB, the results of the fractional 10 Gy IR group were significantly different from the control group; however, the results of the singular 5 Gy IR group were more tended to the control group (Figure 6B). Also, the results of SOD, GSH-px, and MDA verified the conclusion above (Figure 6C).

p53 is strictly regulated by DNA damage and associated with oxidative stress. Therefore, we analyzed p53 and apoptosis-related genes (bax, bcl2, and caspase3) by TPM quantification. The results indicated that IR exposure up-regulated apoptosis-related genes, which may play a major role in IR-induced brain damage (Figure 6D). Additionally, the results were verified by WB (Figure 6E). The results showed that both IR groups had significant differences in p53 and bcl2; the fractional 10 Gy exposure group showed significant up-regulation of bax, while the singular 5 Gy exposure group did not.

Above these results, IR-induced brain injury is mainly linked to DNA damage, oxidative stress, and apoptosis. Due to the results of singular 5 Gy IR exposure, it seems that the organism can repair in a time-dependent manner to some extent without drug intervention two months after single IR exposure.

### 2.7. Singular IR Exposure and Fractional IR Exposure Induce Bone Loss and Promote Osteoclast Maturation in Mice

IR exposure-related low bone mass and strength are associated with increased osteoclast numbers and decreased osteoblast numbers. We also explored the delayed response and damage to femoral tissue by IR exposure. As determined by micro-CT, trabecular bone volume (bone volume per tissue volume [BV/TV]) and 3D bone mineral density (BMD) decreased at the femur in the fractional 10 Gy IR exposure group. This decrease was associated with a decrease in trabecular number and an increase in trabecular spacing, while no changes were detected in the trabecular thickness of the femur. The singular exposure caused a decrease in trabecular number and an increase in trabecular spacing of the femur, while it had no effect on trabecular thickness (Figure 7A).

To further investigate the possible causes of the decrease in bone density caused by IR exposure, we carried out HE staining and Trap staining of mouse femur tissue. The results of HE staining showed that the distal femur developed a cavity structure showing vacuolar structure, especially in the fractional 10 Gy IR exposure group. Trap staining aims to stain osteoclasts in the bone tissue, and the stained osteoclasts appear violet-red. The results of the Trap staining showed that there was almost no positive area in the control group, while the two IR groups, especially the fractional 10 Gy IR group, showed a clear positive reaction at the cancellous bone edge of the distal femur. The number of osteoclasts was counted and found to be significantly higher in the fractional 10 Gy IR exposure group (Figure 7B,C).

### 2.8. IR Exposure Induced Oxidative Stress and Apoptosis in Mouse Femur Tissue

According to WB results, the expression of oxidative stress-related proteins (Nrf2 and HO-1) was reduced and significantly different in the femoral tissues of both IR groups compared to the control group. In addition, the expression of apoptosis-related proteins (p53, bax, bcl-2, and p21) was increased. The expression of bone-related marker proteins (RUNX2, sclerostin, and β-catenin) also verified the staining results of the pathological sections. The expression of osteogenesis-related proteins was reduced in the radiation group (Figure 8A,B).

The results of these experiments indicate that IR exposure causes an increase in mature osteoclasts and a decrease in bone mineral density in the femur and leads to oxidative stress and apoptosis in mouse femur tissue, while the exact type of apoptosis needs to be explored in subsequent experiments (Figure 8C).

## 3. Discussion

The high risk of fractures related to delayed reaction after IR exposure represents a major clinical problem. As we know, TBI-induced injury is often not a single organ damage and often results in a series of adverse reactions. These reactions are delayed in nature and may still be damaging to human health 1–2 months after IR exposure. Thus, in this study, we examined the injury in mice with singular and fractional IR exposure to different doses two months after IR exposure. Our data indicate that the mice showed slow movement, weight loss, and whitening of fur. From the results of the behavior test, two months after IR exposure, mice still show reduced spatial memory and activity, especially in the fractional 10 Gy IR exposure group. This impairment of cognitive function is strongly associated with an increased number of apoptotic neurons in the CA1 region of the hippocampal tissue of mice. Apart from this, bone damage due to IR exposure also plays an important role in clinical treatment. So, in this study, we perform micro-CT scanning, and the results show a significant decrease in BMD, BS, and [BV/TV] in the fractional 10 Gy IR exposure group compared with the control group. According to HE staining and Trap staining, IR-induced bone loss may be linked to the increased number of osteoclasts. The decrease in bone density due to the increase in osteoclast number may be relative to the down-regulation of RUNX2, sclerostin, and beta-catenin. Meanwhile, the brain injury caused by IR exposure is mainly linked to the down-regulation of BNDF and Tau. Above all, the mechanism of cognitive and activity damage was mainly related to oxidative stress and apoptosis induced by DNA damage.

Clinical studies have shown that delayed reaction after IR exposure usually occurs 1–2 months after radiotherapy, and some patients may still have relevant symptoms even 1 year or more after radiotherapy. Due to the diversity and uncertainty of delayed reactions, which means multi-target and multi-organ damage and different times of occurrence, it is still an urgent problem to be overcome in clinical post-radiotherapy treatment. Some studies have reported that delayed reactions after local or systemic radiotherapy, mainly in the neck, head, and bones, lead to apoptosis and neuron inflammation in the brain, resulting in irreversible damage and, in severe cases, mood changes and cognitive dysfunction [30,31]. In the bone tissue, there is a decrease in motor ability and osteoporosis, especially in adolescents.

In recent years, radiotherapy has become an important method to treat cancer and prolong human life. Among them, whole-brain radiotherapy is the most effective method for the treatment of brain cancer. Clinical data show that whole-brain radiotherapy can directly kill cancer cells in brain tissue, prolong the survival time of tumor patients, and inhibit the growth of cancer cells [32,33]. However, at the same time, whole-brain radiotherapy can also damage the normal cells of patients and even damage memory, emotion, cognition, and other functions [34]. Now, more and more people have begun to pay attention to the safety of whole-brain radiotherapy [35]. Radiation-induced bone loss is mainly related to total body radiotherapy and has a high incidence in the elderly. Clinical data show that patients with bone metastases have a suitable survival rate after systemic radiotherapy, but they are still accompanied by adverse reactions such as myelosuppression, lower limb numbness, and mobility difficulties, and most of them are localized adverse reactions [36]. Also, studies have shown that daily intake of appropriate calcium can help improve bone quality [37,38], while other studies have shown that daily calcium intake is not a decisive factor in bone mineral density in the elderly [39]. In this study, both two IR groups still caused bone damage after 60 days; however, whether a high-calcium diet can improve this situation will be discussed in future studies.

Therefore, it is very necessary to study radiation damage. In the selection of experimental animals, most of the studies related to radiation studies mostly use rodents for experiments [40,41,42,43]. Because mice are about twice as sensitive to radiation as humans, obvious radiation damage can be observed at lower doses. There is no significant difference in radiation sensitivity among different strains of mice. However, BALB/c mice are more sensitive to radiation at low doses. This is because BALB/c mice are particularly sensitive to the effects of radiation due to an unknown autosomal recessive genetic locus and cannot tolerate the same radiation dose as C57BL/6 mice [44,45]. In addition, other animal models also have been used to evaluate radiation-related injuries and diseases. For example, the cornea irradiated by gamma rays for corneal transplantation was evaluated in a rabbit model, and the transparency of the graft, corneal neovascularization, and edema were compared [46,47,48] in addition to the study of radiation-induced skin injury and lung injury in domestic pig models [49].

Radiotherapy, as a major method of cancer treatment, is based on the fact that ionizing radiation can directly cause DNA damage to cancer cells, thereby inhibiting their DNA replication, leading to cell cycle arrest, and thus inhibiting their proliferation [50]. IR can cause DNA double-strand breaks in cells either directly or indirectly through the production of break-inducing reactive oxygen species. Without repair, DNA-damaged cells may undergo multiple modes of cell death, including apoptosis, mitotic catastrophe, and autophagy, or enter a state of stunted growth, even leading to cellular senescence [51,52,53]. DNA damage promotes senescence in both cancer and normal cells. Senescent cells secrete a range of pro-inflammatory factors, collectively known as senescence-associated secretory phenotypes (SASP). SASP promotes tumor cell proliferation and immunosuppression, creating an even more severe vicious circle [54,55,56]. However, regardless of whether the radiation is applied to the whole body or locally, ionizing radiation not only causes damage to cancer cells but also to healthy cells, and DNA damage is a direct response to radiation-induced damage, leading to oxidative stress and apoptosis. P53 is a transcription factor activated by DNA damage, which has received widespread attention as a crossroads of multiple pathways, including oxidative stress, cell cycle regulation, and apoptosis. Inhibition of p53 activity has shown positive clinical effects in cancer and senescence.

However, the role of p53 in radiation-induced cognitive function and bone tissue damage is unclear. As shown in Figure 9, p53, as an upstream transcription factor, can regulate oxidative stress in organisms depending on the degree of stress, which can regulate both the reactive oxygen species and antioxidant enzyme system directly and then act on p21-Nrf2 to regulate the antioxidant system indirectly [55,57,58,59]. In addition, p53 can also regulate apoptosis by regulating the secretion of BH3 and indirectly regulating the expression of bax and bcl2 [60]. The results of this experimental study indicate that IR exposure as a stressor directly causes DNA damage in organisms, activates a substantial increase in p53 expression, and induces oxidative stress damage and apoptosis in organisms. In the Wb results of bone and brain tissues, it can be seen that singular or fractional IR exposure results in a significant elevation of p53, p21, and bax expression and a decrease in Nrf2 and bcl2 expression. These two regulatory mechanisms acted synergistically in vivo to promote decreased secretion of neurotrophic factors in mouse brain tissue, elevated number of osteoclasts in mouse femur tissue, and markedly decreased bone density, which led to decreased cognitive function and decreased motor ability in mice after singular or fractional IR exposure.

Taken together, in this study, we compared the differences in behavior tests, pathological analyses, transcriptome sequencing, and molecular biological validation between singular 5 Gy IR exposure and fractional 10 Gy IR exposure. We found that the cognitive-motor function damage caused by singular 5 Gy IR exposure was unstable, especially in the aspects of activity and bone damage, and our experiments indicated that fractional 10 Gy TBI was a stable model of multi-organ damage for the study of radiation mechanism and the screening of anti-radiation drugs. Increased evidence shows that traditional Chinese medicine (TCM) played a very suitable anti-radiation effect [61]. Studies have shown that Lycium barbarum extract has a neuroprotective effect on radiation-induced neurobehavioral changes such as cognitive impairment and depression [62]. From the point of view of antioxidation, many TCMs may play a better role, such as Cistanche deserticola and Houttuynia cordata [63,64]. In the future, the long-term anti-radiation effect of TCM can be studied using this model. The reason for the instability of these indexes may be that the singular 5 Gy IR exposure is reversible for the damage of the antioxidant system, which means the antioxidant enzyme system is still able to play its role after two months; however, the above speculation has not been verified yet, and we will carry out further experiments to improve our understanding of singular or fractional IR research in future. In addition, further experiments will be conducted in the future to explore the damage to the antioxidant enzyme system by singular or fractional IR exposure, to enrich the role of the p53 gene in this process, and to provide new targets and therapeutic ideas for the treatment of delayed post-irradiation injury.

## 4. Materials and Methods

### 4.1. Chemicals and Reagents

Assay kits for superoxide dismutase (SOD) were purchased from Nanjing Jiancheng Bioengineering Institute (Nanjing, China), mouse malonaldehyde (MDA) and mouse glutathione peroxidase (GSH-px) ELISA kits were purchased from Beijing Chengzhikewei Biotechnology Co., Ltd. (Beijing, China), and radioimmunoprecipitation assay (RIPA) lysis buffer and the BCA protein detection kit were purchased from Epizyme Biotech (Shanghai, China). Antibodies against nuclear factor E2-related factor 2 (Nrf2, 1:1000), heme oxygenase-1 (HO-1, 1:1000), p21 (1:1000), p53 (1:1000), Tau (1:1000), BDNF (1:1000), GAPDH (1:1500), β-actin (1:1500), and anti-rabbit horseradish peroxidase (HRP)-conjugated IgG were purchased from Cell Signal Technique (Boston, MA, USA). Antibodies against RUNX2 (1:1000), sclerostin (1:1000), and β-catenin (1:1000) were purchased from Abcam (Cambridge, UK).

### 4.2. Animal Model

Considering that male mice are more sensitive to radiation, we selected male C57BL/6J mice for this experiment [65,66]. Male C57BL/6J mice weighing 20 ± 2 g were purchased from Beijing Weitong Lihua Experimental Animal Technology, License No: SCK (Beijing) 2021-006. Mice were housed in a barrier feeding system at the Experimental Animal Center of the Military Medical Research Institute, with 5 animals per cage and free access to food and water (standard solid feed and sterilized tap water provided by the animal center). All mice experiments were performed with the approval of the intramural Committee on Ethics Conduct of Animal Studies of the Academy of Military Medical Sciences, China, ethics No. IACUC-DWZX-2020-762.

Animals were randomly divided into three groups (n = 7 per group): control group (CON), singular 5 Gy IR exposure group (5 Gy), and fractional 10 Gy IR exposure group (5 Gy*2). Fractionation doses are doses in which the total dose of radiation is divided into several smaller doses over a period of several days/weeks. In our study, we focused on the 60 days after the first IR exposure, so the fractional 10 Gy group means two months of fractional 10 Gy IR exposure. Both IR exposure groups were fixed in irradiation-specific boxes and received total body irradiation (TBI) at a distance of 2 m from the irradiation source at a dose rate of 1.18 Gy/min. The fractional 10 Gy IR exposure group received a second TBI 30 days after singular 5 Gy IR exposure, with the same operation as the first IR exposure. After irradiation, the mice were placed back into the animal center barrier housing system for further housing. The control group was also fixed in the irradiation-specific boxes with no irradiation. As shown in Figure 1 the behavioral tests were performed sequentially, and then we euthanized the mice and collected the brain and femur for subsequent research (Figure 10).

### 4.3. Behavior Test

Open field test (OFT): To reflect the spontaneous activity behavior of experimental animals, open field tests were recorded for 5 min after 1 min of acclimatization. During the test period, the experimental box was cleaned with 75% alcohol before proceeding to the next mouse.

Water maze test (WMT): The water maze test is the classic behavior test to investigate changes in spatial memory ability. Each mouse was trained 4 times a day for 4 days. Prior to the first experiment, each mouse was placed on a platform for 15 s and then allowed to swim freely for 30 s before being helped onto the platform for another 15 s of rest. In each trial, in one of the four quadrants, the mice were placed in the water facing the marker for that quadrant, and the time it took to release the mice to find the hidden platform was recorded. Mice that successfully found the platform were allowed to remain on the platform for 15 s. Mice that did not find the platform within 60 s were placed on the platform for 15 s at the end of the experiment. The platform was removed on day 5, and each mouse was allowed to swim for 60 s for the probe test. The time the mice spent swimming in the target quadrant and the three non-target quadrants was measured separately. For probe trials, the number of crossings of the platform site was measured and counted, and all data were recorded using a computerized video recording system.

### 4.4. Microcomputed Tomography (μCT)

Mouse femoral tissues were similarly fixed in fixed in 4% paraformaldehyde and then scanned using Bruker Micro-CT Skyscan 1276 system (Kontich, Belgium). Scan settings are as follows: voxel size 10.033633/6.533712 μm, medium resolution, 85 kV, 200 μA, 1 mm Al filter, and integration time 384 ms. Density measurements were calibrated to the manufacturer’s calcium hydroxyapatite (CaHA) phantom. Analysis was performed using the manufacturer’s evaluation software. Reconstruction was accomplished by NRecon (version 1.7.4.2.2. Three-dimensional images were obtained from contoured 2D images by methods based on distance transformation of the grayscale original images (Ctvox; version 3.3.0). The 3D and 2D analyses were performed using the software CT Analyser (version 1.20.3.0).

### 4.5. Pathological Analysis

Mouse brain tissue was collected after sampling, fixed in 4% paraformaldehyde, dehydrated and embedded in paraffin, and subsequently prepared in paraffin sections of approximately 5 μm thickness, which required dewaxing and rehydration prior to staining. Mouse femoral tissues were similarly fixed in 4% paraformaldehyde and then dehydrated and filmed for later staining after performing micro-CT scanning experiments.

Hematoxylin–eosin staining: Paraffin sections were dewaxed to water and then stained with hematoxylin. The sections were stained in hematoxylin staining solution for 1–2 min, washed in tap water, divided in differentiation solution, washed in tap water, returned to blue in return blue solution, and rinsed in running water. After completion, the sections were stained with eosin staining solution for 2–3 min. Finally, the sections were dehydrated and sealed.

Nissl staining: Paraffin sections were dewaxed to water, and then toluidine blue-stained animal tissue sections were put into the staining solution for 5 min, washed with water, slightly differentiated by 1% glacial acetic acid, and the reaction was terminated by tap water washing, and the degree of differentiation was controlled under the microscope, and after tap water washing, the sections were placed in the oven to dry. The slices were put into clean xylene transparent for 5 min, and neutral resin sealed the slices. Finally, a microscopic examination was performed, and images were collected for analysis.

Trap staining: Paraffin sections were dewaxed to water for Trap staining, sections were drawn in circles with histochemistry, incubated dropwise with TRAP incubation solution placed at 37 °C for 1–2 h, washed with distilled water, and osteoblasts were observed microscopically in burgundy color. After Trap staining was finished, cell nuclei were stained with hematoxylin, sections were stained with hematoxylin for 1–3 min, washed with tap water, aqueous hydrochloric acid solution for a few seconds, rinsed with tap water, aqueous ammonia solution returned to blue, and running water rinsing. After staining, the sections were dehydrated with xylene, dried, and sealed with neutral gum. Finally, microscopic examination and image acquisition were performed for analysis.

### 4.6. ELISA

Tissues were taken from the mouse cortical region and weighed, homogenized in phosphate buffer (1:10), and the supernatant was collected by centrifugation (4 °C, 5000× *g*, 15 min). Measurements of SOD GSH-px and MDA were according to the instructions.

### 4.7. RNA Preparation and Sequencing

A total amount of 2 μg RNA per sample was used as input material for the RNA sample preparations. Sequencing libraries were generated using NEBNext^®^ Ultra™ RNA Library Prep Kit for Illumina^®^ (#E7530L, Ageilent Technologies, CA, USA) following the manufacturer’s recommendations, and index codes were added to attribute sequences to each sample. RNA concentration of the library was measured using Qubit^®^ RNA Assay Kit in Qubit^®^ 3.0 (Ageilent Bioanalyer 2100 system, Ageilent Technologies, CA, USA) to preliminary quantify and then dilute to 1 ng/μL. The clustering of the index-coded samples was performed on a cBot cluster generation system using HiSeq PE Cluster Kit v4-cBot-HS (Illumina, San Diego, CA, USA) according to the manufacturer’s instructions. After cluster generation, the libraries were sequenced on an Illumina platform, and 150 bp paired-end reads were generated.

### 4.8. Western Blot

Total protein extracts were prepared in 200μL of ice-cold RIPA lysis buffer, combining 30 mL of 10 mg/mL PMSF solution, 30 mL of Na3VO4, and 30 mL of protease inhibitors cocktail per gram of tissue. Lysates were centrifuged at 10,000 rpm for 10 min at 4 °C, and then the supernatants were removed and centrifuged again. The supernatants were collected. Protein levels in the supernatants were determined using the BCA assay kit. Samples (40 mg each) were separated by denaturing SDS-PAGE and transferred to a PVDF membrane by electrophoretic transfer. The membrane was pre-blocked with 5% non-fat milk in Tris-buffered saline (TBS) and incubated overnight with primary antibodies (1:1000). Quantitation of detected bands was performed with ImageJ version 1.0.8 (ImageJ, NIH IMAGE, USA). Each density was normalized using each corresponding beta-actin density as an internal control, and we standardized the density of vehicle control for relative comparison as 1.0 to compare other groups.

### 4.9. Statistics

All the data are presented as mean ± SD. Statistical analyses were performed on GraphPad Prism version 9 (GraphPad Software, La Jolla, CA, USA). Statistical significance was calculated with a one-way ANOVA corrected for multiple comparisons, which in all figures are represented by: * *p* < 0.05; ** *p* < 0.01; *** *p* < 0.001.

## 5. Conclusions

Singular or fractional IR exposure led to a decrease in spatial memory ability and activity in mice, and the cognitive and motor functions gradually recovered after singular 5 Gy IR in a time-dependent manner, while the fractional 10 Gy IR group could not recover. The decrease in bone density due to the increase in osteoclast number may be relative to the down-regulation of RUNX2, sclerostin, and beta-catenin. Meanwhile, the brain injury caused by IR exposure is mainly linked to the down-regulation of BNDF and Tau. Above all, the mechanism of cognitive and activity damage was mainly related to oxidative stress and apoptosis induced by DNA damage. The damage caused by fractional 10 Gy TBI is relatively stable and can be used as a stable multi-organ injury model for radiation mechanism research and anti-radiation medicine screening.

## Figures and Tables

**Figure 1 ijms-25-00337-f001:**
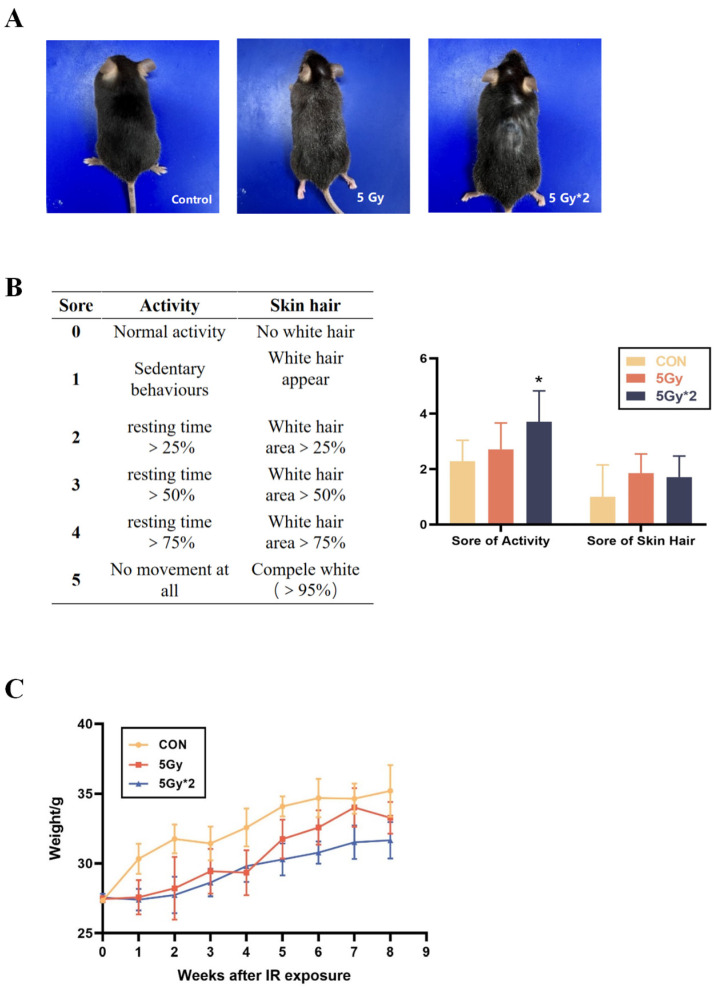
Physiological change in singular 5 Gy and fractionated 10 Gy IR exposure. (**A**) Change in mice appearance. (**B**) Score of activity and hair (*n* = 5, IR/CON). (**C**) Body weight change curves of different groups of mice (IR/CON). * *p* < 0.05 vs. CON.

**Figure 2 ijms-25-00337-f002:**
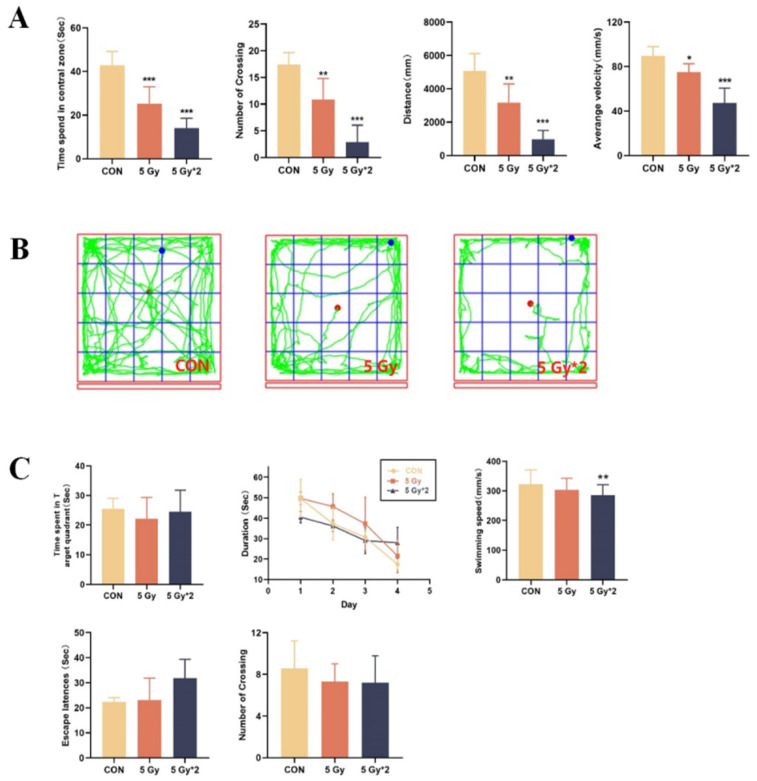
Open field tests were recorded for 5 min after 1 min of acclimatization. (**A**) Experimental indicators of open field test: time spent in the central area, number of crossing, travel distance, average velocity. (**B**) Orbit of open field test; The water maze test is to investigate changes in spatial memory ability. Each mouse was trained 4 times a day for 4 days, and in the probe test, the platform was removed, and each mouse was allowed to swim for 60 s. (**C**) Experimental indicators of Morris water maze test: time spent in the target quadrant, duration to platform during cued training, swimming speed, escape latency, and number of platform crossings. * *p* < 0.05, ** *p* < 0.01, *** *p* < 0.001 vs. CON.

**Figure 3 ijms-25-00337-f003:**
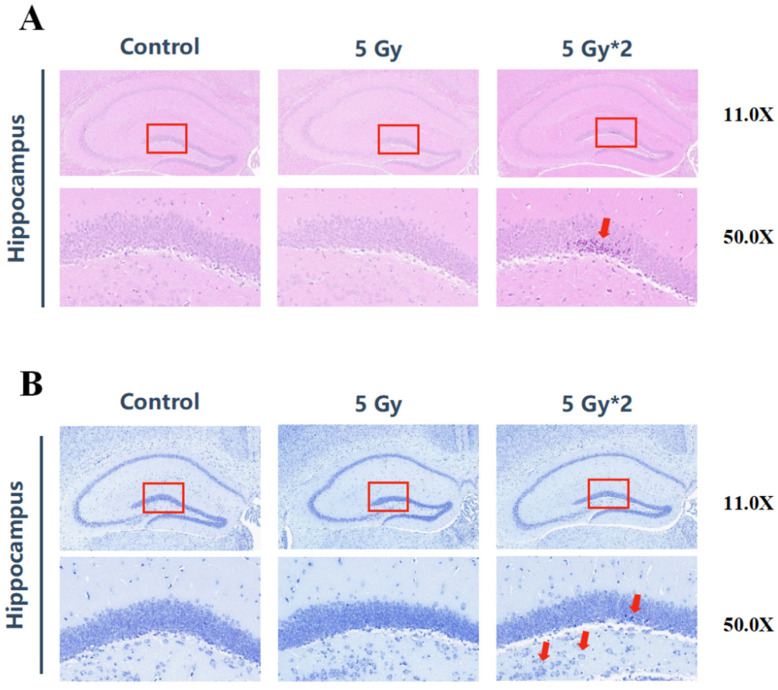
IR exposure induced brain tissue damage in mice. The red square represents the DG region of the hippocampus. (**A**) H&E staining of brain tissue (*n =* 4) in the hippocampal DG region (50×). The arrows indicated neuronal necrosis. There were significant changes between 5 Gy*2 and CON (10 Gy IR/CON). (**B**) Nissl staining of brain tissue (*n =* 4) in the hippocampal DG region. The arrows indicated swollen neuronal cells with apoptotic vesicles (50×). Compared with CON, 5 Gy*2 showed more serious damage in the swelling of neuronal cells (10 Gy IR/CON).

**Figure 4 ijms-25-00337-f004:**
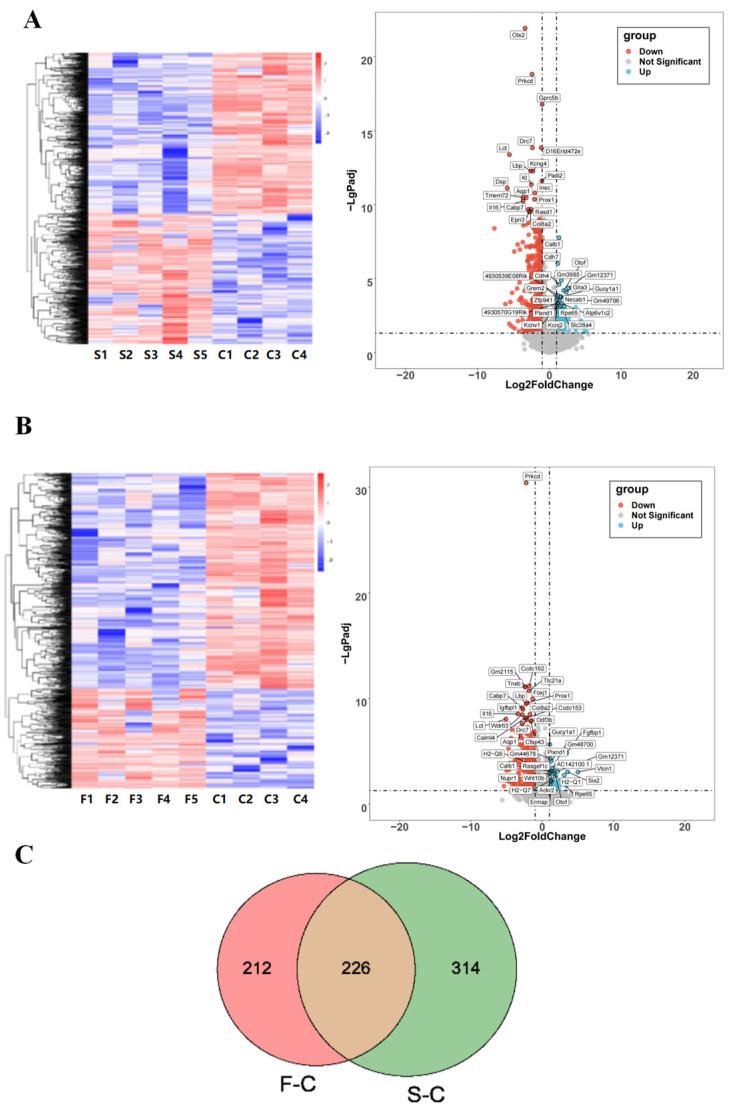
Effects of radiation on the transcriptome of mouse hippocampal tissue. (**A**) CON vs. singular 5 Gy IR exposure group heat map and volcano map. (**B**) CON vs. fractional 10 Gy IR exposure group heat map and volcano map. Heat map showed that both IR groups had a suitable gene separation compared with the CON group (IR/CON). In the volcano map, red represented down-regulated genes and blue represented up-regulated genes. (**C**) Venn diagram comparing the significant DEGs in each group. The green part represented DGEs of 5 Gy-CON (5 Gy/CON), and the red part represented DGEs of 5 Gy*2-CON (5 Gy*2/CON). (**D**) F/C and S/C top 20 GO enriched terms of 226 key overlapping DEGs. Left was up-regulation of DEGs; right was down-regulation of DEGs (*n =* 4–5). From top to bottom are molecular function, biology progress, and cellular components. The level of gene expression is shown by the odds ratio.

**Figure 5 ijms-25-00337-f005:**
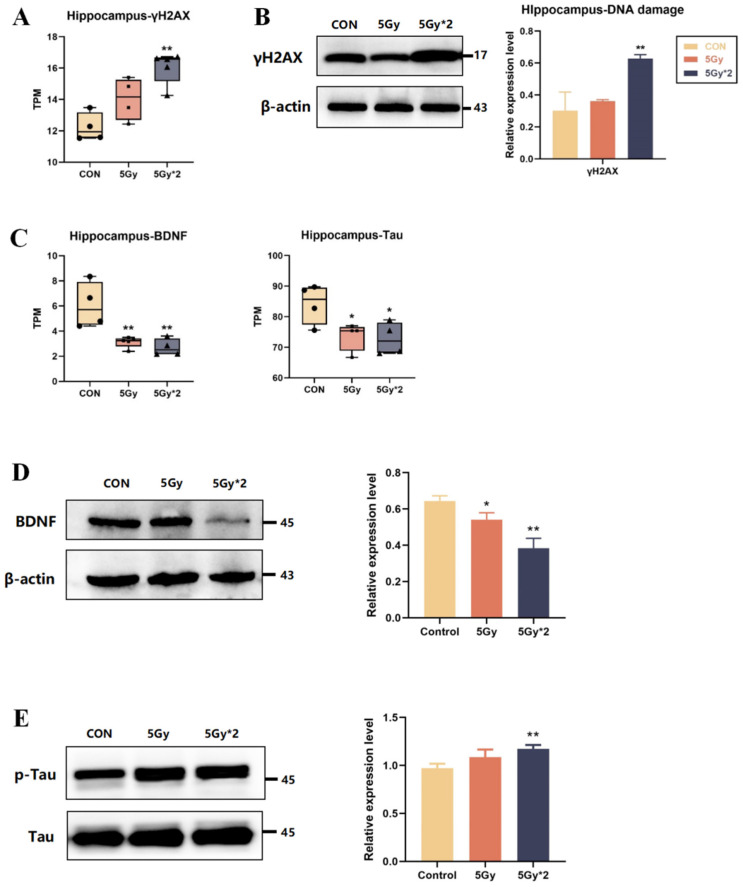
The effect of IR exposure on DNA damage and cognition-related marks in mouse hippocampal tissue. (**A**) Boxplots of transcripts per million transcripts (TPM) of marker of DNA damage gene: γH2AX (IR/CON). (**B**) Western blot analysis of γH2AX in hippocampal tissue. Both IR groups showed significantly elevated (IR/CON). (**C**) Boxplots of transcripts per TPM of cognition-related genes for each group: BDNF and Tau (IR/CON). (**D**) Western blot analysis of cognition-related protein in hippocampal tissue. The content of BDNF significantly decreased (5 Gy*2/CON). (**E**) Western blot analysis of p-Tau and Tau in hippocampal tissue. The result of p-Tau/Tau showed a significant increase (5 Gy*2/CON). Data are presented as mean ± SE, *n =* 4–5, * *p* < 0.05, ** *p* < 0.01 vs. CON.

**Figure 6 ijms-25-00337-f006:**
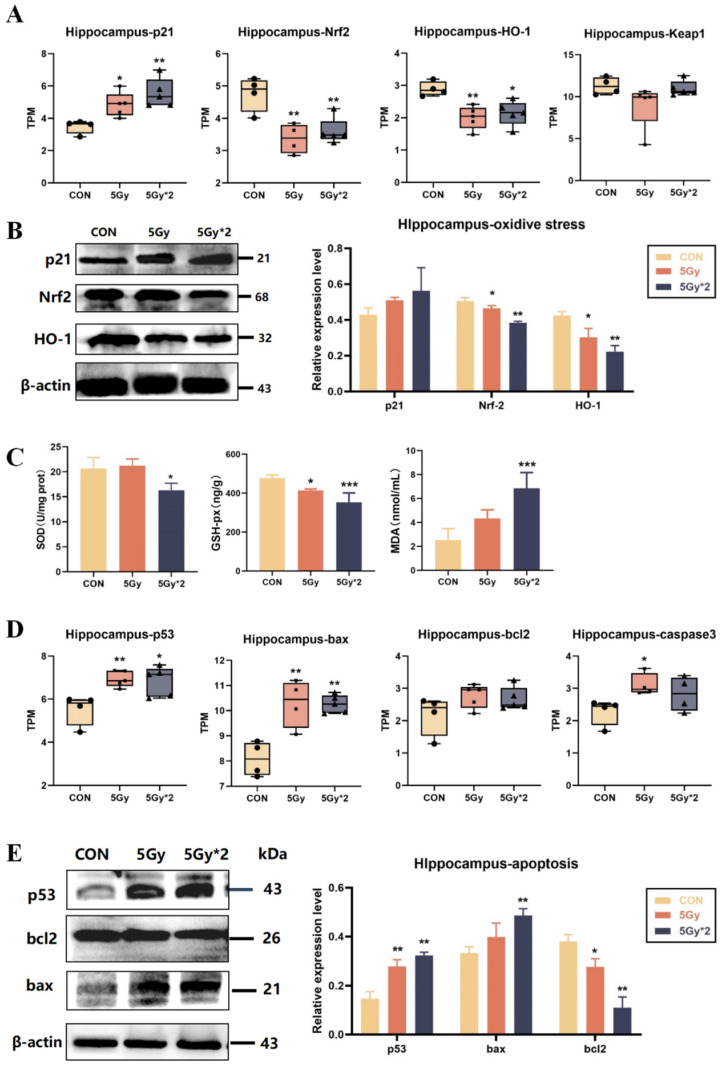
IR exposure increases oxidative stress and apoptotic response in mouse hippocampal tissue. (**A**) Boxplots of transcripts per TPM of oxidative stress-related genes for each group: p21, Nrf2, and HO-1 (IR/CON). (**B**) Western blot analysis of oxidative stress-related protein in hippocampal tissue. (**C**) Activity of SOD, MDA, and GSH-px in mice cortex. SOD = superoxide dismutase. Compared with CON, the content of antioxidant enzymes decreased significantly, and the content of peroxidase increased significantly (5 Gy*2/CON). (**D**) Boxplots of transcripts per million transcripts (TPM) apoptosis-related genes: p53, bax, and bcl2 (IR/CON). (**E**) Western blot analysis of apoptosis-related protein in hippocampal tissue (IR/CON). Data are presented as mean ± SE, *n =* 4–5, * *p* < 0.05, ** *p* < 0.01, *** *p* < 0.001 vs. CON.

**Figure 7 ijms-25-00337-f007:**
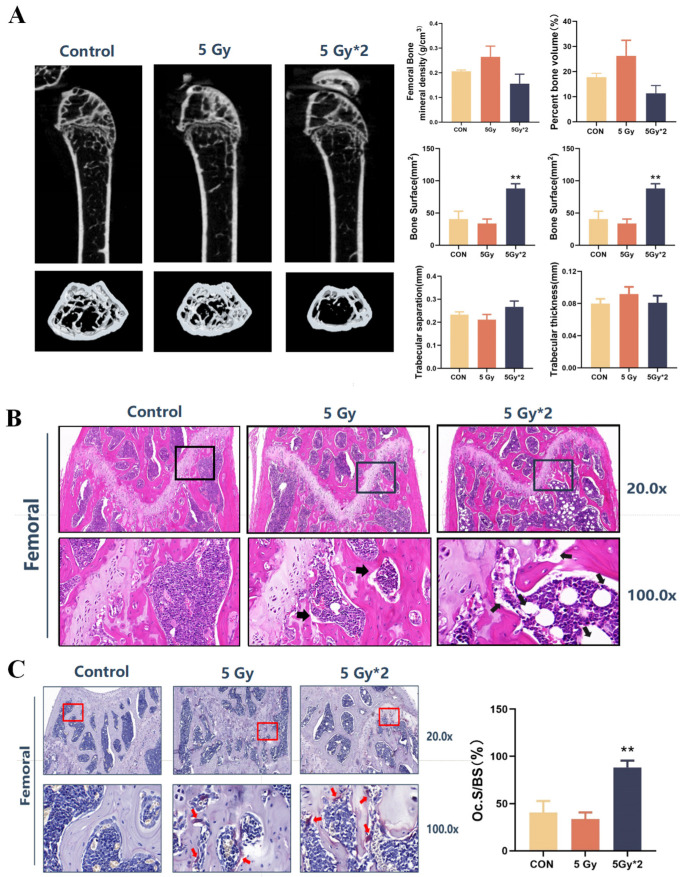
IR exposure induced bone loss and promoted osteoclast maturation in mice. The square region represents the cartilage joint of the distal femur. (**A**) Imaging and quantification of femur bones from CON and IR-exposed mice by micro-CT. A total of 60 days later, IR exposure still caused trabecular bone volume and 3D bone mineral density (BMD) to decrease (*n =* 4, IR/CON). (**B**) H&E staining of femur bones (*n =* 4) (20× and 100×). The arrows indicated the cavity of the femur. The distal femur in the 5 Gy*2 group developed a cavity structure showing vacuolar structure (5 Gy*2/CON). (**C**) TRAP staining of femur bones. The arrows indicated TRAPase activity in femur bones (20× and 100×). The number of osteoclasts was counted and found to be significantly higher in the 5 Gy*2 group (5 Gy*2/CON). Data are presented as mean ± SE, *n =* 4–5, ** *p* < 0.01 vs. CON.

**Figure 8 ijms-25-00337-f008:**
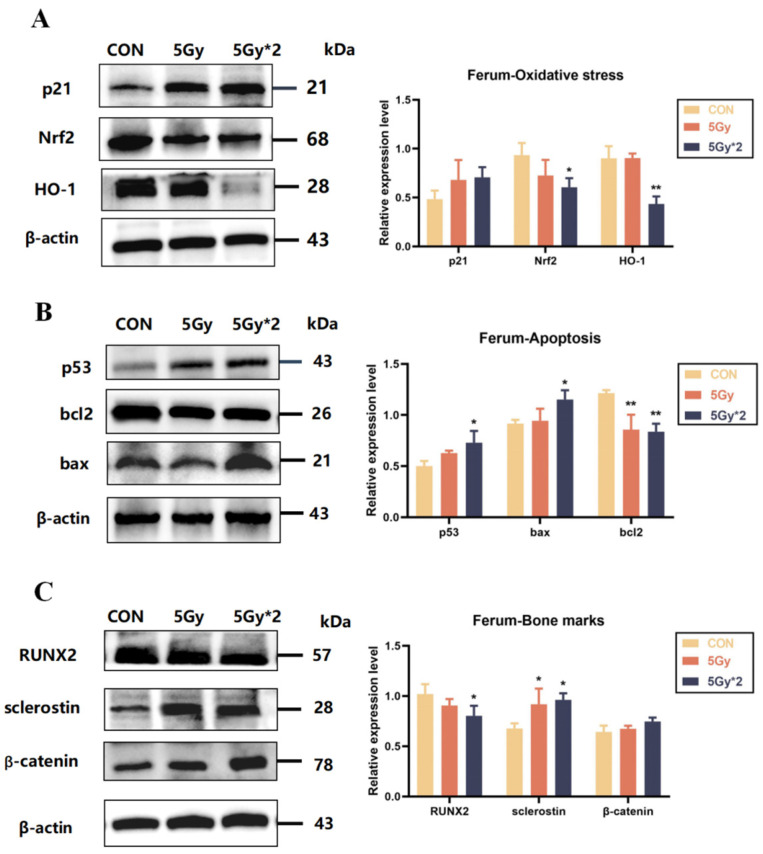
IR exposure increases oxidative stress and apoptotic response in mice femur bones. (**A**) Western blot analysis of oxidative stress-related protein in femur bones: p21, Nrf2, and HO-1 (IR/CON). (**B**) Western blot analysis of apoptosis-related protein in femur bones: p53, bax, and bcl2 (IR/CON). (**C**) Western blot analysis of bone loss-related protein in femur bones: RUNX2, sclerostin, and β-catenin (IR/CON). Data are presented as mean ± SE, *n =* 4–5, * *p* < 0.05, ** *p* < 0.01 vs. CON.

**Figure 9 ijms-25-00337-f009:**
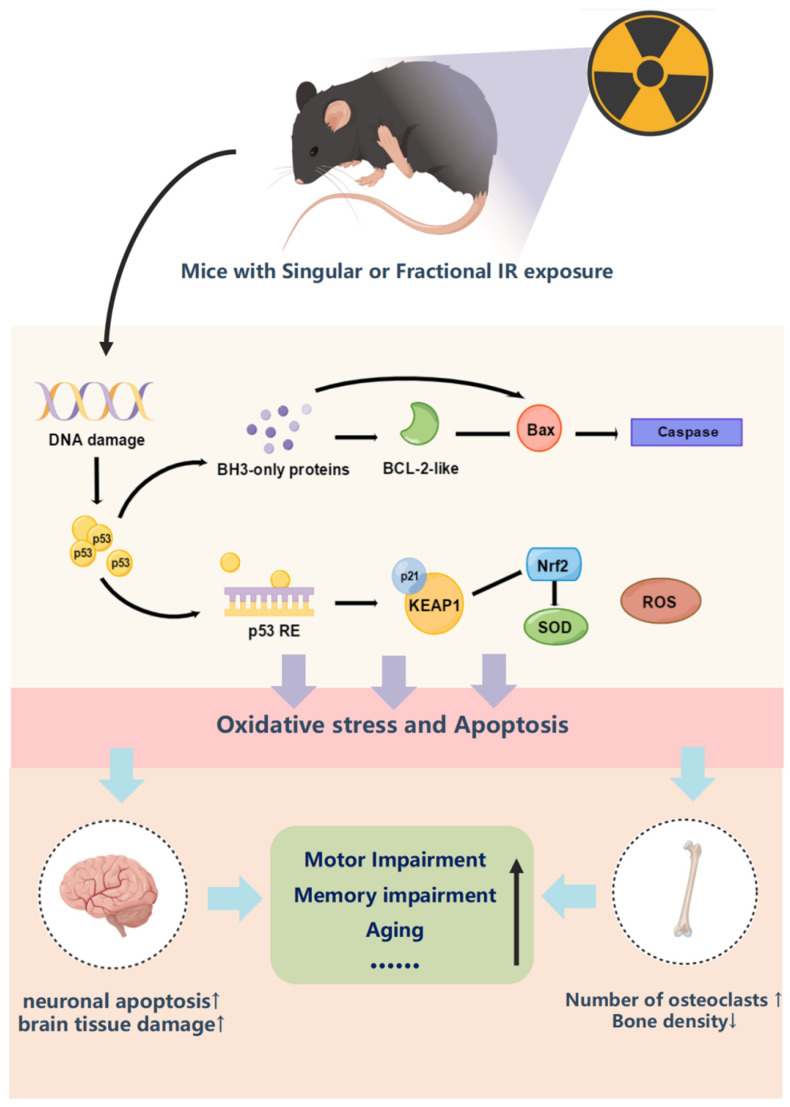
Mechanism of p53 regulation of oxidative stress and apoptosis and pathological characterization results.

**Figure 10 ijms-25-00337-f010:**
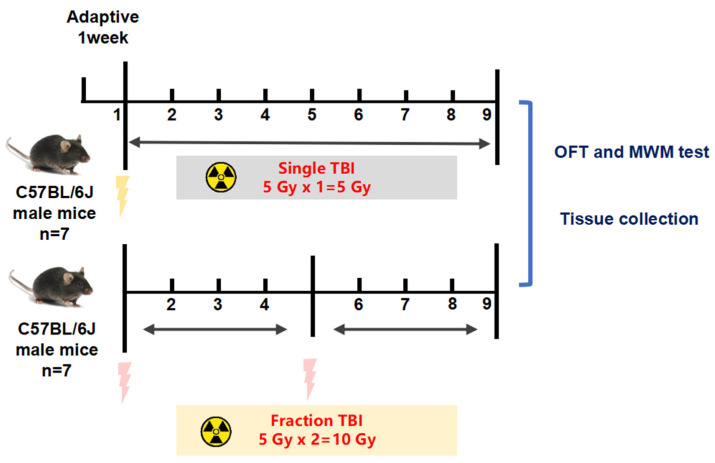
Study design and animal treatments. Animals were randomly divided into three groups (*n = 7*): control group (CON), singular 5 Gy IR-exposed group (5 Gy), and fractional 10 Gy IR-exposed group (5 Gy*2). Both IR exposure groups received total body irradiation (TBI). The fractional 10 Gy IR exposure group received a second TBI 30 days after singular 5 Gy IR exposure.

## Data Availability

Data are contained within the article.

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
