# Peer review of "Delayed Reaction of Radiation on the Central Nervous System and Bone System in C57BL/6J Mice"

_ijms, 2023, doi:10.3390/ijms25010337_

Round 1
Reviewer 1 Report
Comments and Suggestions for Authors
COMMENTS FOR THE AUTHORS:
The authors of the manuscript “Risk and adaptive response to radiation on the central nervous system and bone system in C57BL/6J mice” establish an animal model of delayed radiation damage, focusing their studies on hippocampus and bone.
They studied mechanisms and outcomes of long-term radiation injury in a controlled environment; the aim of the study was to provide a mouse model suitable for developing therapeutic interventions that could protect from radiation injury or enhance their recovery following exposure.
Protection from radiotherapy side effects his is a relevant problem to analyse in fact, there are lots of clinical cases of delayed reactions after radiotherapy, this poses many new challenges to the quality of life of cancer patients, and this mouse model provides a useful starting point.
Although this paper is potentially interesting and adds useful data to the specific field, substantial improvements are needed before publication in International Journal of Molecular Sciences.
Major points
1. The title does not fit well for the paper; the authors mentioned adaptive response, but they didn’t clarify the concept in the main text.
2. Abstract and main text: the use of term Dose fractionation is inappropriate.
Fractionation doses are doses, in which the total dose of radiation is divided into several, smaller doses over a period of several days/weeks, the fractionation generally provokes fewer toxic effects on healthy cells. The authors divided the time of exposition not the dose, in fact they supplied 10 Gy of X Ray in two administrations. The general meaning is completely different, because it is well-known the dose-dependent effect. The novelty of the paper consists of the study of long-term radiation injury effects and the target tissue bone.
3. Extensive editing of the text is required, not only for English language. Considering what underlined above, the test should be extensively and carefully revised; in paragraphs of results, some details should be added to explain the research design and methods.
4. Many of the cited references are not relevant to the research.
Minor points
1. Original images file should contain complete hybridized membranes.
2. Figure 1: typing error mice scarification
3. Figure 2 A Picture 3 Gy 4 months is not relevant to research and not mentioned in the results.
4. Figures legend 2 ..exposure on physiological.. is inappropriate.
5. Figures legends and text contain repetitions, technical details should be included in legend to better clarify the results obtained.
6. In the text: In the probe trial, there was significant difference in spatial memory ability between the control group and the Fractional 10Gy IR exposure group in swimming speed and escape latences. (Figure 3C). Figure doesn’t support statement.
7.
In the text: However, Fractional 10Gy IR exposure group was observed to have varying degrees of abnormalities such as loosely arranged pyramidal cells, crinkled neurons, deepened cell staining, and poorly demarcated nuclei in CA1 region (Figure 4A). According to Nissl staining, both IR expo-sure groups showed swelling of neuronal cells, neuronal necrosis and nucleolytic in CA1 region, especially in the Fractional 10 Gy IR exposure group (Figure 4B).
In the text: Figure 4. IR exposure induced brain tissue damage in mice. A. H&E staining of brain tissue in the hippocampal CA1 region (50X). The arrows indicated neuronal necrosis; B. Nissl staining of brain tissue in the hippocampal CA1 region. The arrows indicated swollen neuronal cells with apoptotic vesicles(50X).
NO CA1 region is that analysed by the Authors, but DG; some pathways had been gotten mixed up.
8 In the text: a total of 438 DEGs in the Fractional 10Gy IR exposure group com-pared to the control group, of which 138 genes were expressed up-regulated and 30 genes were expressed down-regulated (Figure 5B). Error typing 300 genes
9 In the text: As shown in Figure 6 C-D, BDNF and Tau genes were significantly reduced in both IR exposure groups and had significant differences with control group Tau; WB Figure doesn’t support the statement, the protein slightly increase.
10 Figure 7A Hippocampus Keap1 wasn’t cited in the text, and 7D caspase 3 too.
11 Figure 9 p21, Nrf-2, RUNX2, sclerostin, b-catenin western blots should be replaced because are of low quality, and densitometric analysis didn’t reflect the quantity of proteins (Con p21 is lower than Nrf2?)
12 The quality of the figures is too low, should be improved.
13 The authors should set the comparison between conditions, IR/CON or CON/IR and the setting should be maintained.

An extensive editing of the text is required, not only for English language.
Reviewer 2 Report
Comments and Suggestions for Authors
This is a report of effects of each of two TBI protocols (either 5Gy TBI once or 5Gy TBI twice delivered 30 days apart, total 10Gy, on male C57Bl/6J mice with respect to brain and bone effects measured at 6o days. The results are interesting and useful.
Major Comments:
Please consider the results with female mice as well. Indicate that they might be different given the radioresistance of females. Fig.1 should be adjusted to indicate that the 2 Fraction group is being measured at 90 days after the first radiation fraction.
The results would be improved by assaying more radiation doses in both the single fraction and two fraction models. The big differences seen between the two treatment groups could have been attributable to the different total dose delivered or to the 30 day rest period between doses.
There should be more discussion about why only Brain and Bone were chosen. Could a different high calcium diet have decreased the prominent bone effect of the two dose model? The brain effect for the two dose model was dramatic in both structural changes and in gene expression and cognitive/motor changes. How are these three data sets linked? What happened to bone marrow, intestine, and lung during these protocols? Why was 60 days chosen? Would animals in the two dose group have improved in brain and bone function if held for a longer interval ? Would the single dose group have shown more severe brain and bone effects if observed earlier. Should be discussed.
Minor Points:
Literature cited is incomplete. Please review more recent studies of brain and bone-both TBI and localized irradiation effects. Also, irradiation effects in other animal models and other mouse strains. Include relevant clinical data from Whole Brain irradiated or bone irradiated patients. Please cite relevant radiation protectors and mitigators which either are being tested or could be tested in this model.
